# Hyperbolic band topology with non-trivial second Chern numbers

Weixuan Zhang [1,2,4], Fengxiao Di[1,2,4], Xingen Zheng [1,2], Houjun Sun [3] & Xiangdong Zhang [1,2] ✉

Topological band theory establishes a standardized framework for classifying different types of topological matters. Recent investigations have shown that hyperbolic lattices in non-Euclidean space can also be characterized by hyperbolic Bloch theorem. This theory promotes the investigation of hyperbolic band topology, where hyperbolic topological band insulators protected by first Chern numbers have been proposed. Here, we report a new finding on the construction of hyperbolic topological band insulators with a vanished first Chern number but a non-trivial second Chern number. Our model possesses the non-abelian translational symmetry of {8,8} hyperbolic tiling. By engineering intercell couplings and onsite potentials of sublattices in each unit cell, the non-trivial bandgaps with quantized second Chern numbers can appear. In experiments, we fabricate two types of finite hyperbolic circuit networks with periodic boundary conditions and partially open boundary conditions to detect hyperbolic topological band insulators. Our work suggests a new way to engineer hyperbolic topological states with higher-order topological invariants.

Topological band theory provides a unified framework for characterizing a wide range of topological states of quantum matters[1-8] and classical wave systems[9-16]. In this theory, band structures of both quantum and classical systems with space-translation symmetries can be classified by topological invariants defined in the momentum space. The pioneering example is the first Chern number (or TKNN invariant) for topological band structures in two-dimensional (2D) Brillouin zone[17-19]. Such a topological invariant plays a key role in characterizing various low-dimensional topological phases, such as the 2D quantum Hall effect, topological insulators and superconductors, and topological semimetals. Except for the first Chern number, the $n$th Chern numbers defined in $2n$-dimensional manifolds can also identify many novel topological states in high dimensions. For example, the second Chern number provides criteria for the appearance of 4D quantum Hall effect[20] and 5D topological semimetals with non-abelian Yang-monopoles or linked Weyl surfaces[21,22]. In much higher dimensions, the 6D

quantum Hall effect is characterized by the third Chern number following a similar extension-method. To date, topological band theory accomplished with different types of topological invariants is mainly focusing on the periodic system in Euclidean space.

On the other hand, hyperbolic lattices, which are regular tessellations in the curved space with a constant negative curvature, have been widely investigated as mathematical objects over past decades[23]. The recent ground-breaking implementation of two-dimensional hyperbolic lattices in circuit quantum electrodynamics[24] and topolectrical circuits[25] has stimulated numerous advances in hyperbolic physics[26-33]. Inspired by the exotic geometric properties of hyperbolic lattices, there are many investigations on the construction of hyperbolic topological states in real space[34-36]. For example, the non-Euclidean analog of the quantum spin Hall effect in hyperbolic lattices has been proposed with a tree-like design of the Landau gauge[34]. In addition, the boundary-dominated hyperbolic Chern insulator has

[1]Key Laboratory of advanced optoelectronic quantum architecture and measurements of Ministry of Education, School of Physics, Beijing Institute of Technology, 100081 Beijing, China. [2]Beijing Key Laboratory of Nanophotonics & Ultrafine Optoelectronic Systems, School of Physics, Beijing Institute of Technology, 100081 Beijing, China. [3]Beijing Key Laboratory of Millimeter Wave and Terahertz Techniques, School of Information and Electronics, Beijing Institute of Technology, Beijing 100081, China. [4]These authors contributed equally: Weixuan Zhang, Fengxiao Di. ✉e-mail: zhangxd@bit.edu.cn

been theoretically proposed and experimentally fulfilled by circuit networks[35]. Those intriguing features of topological states are probes on illustrating non-Euclidean topology, and suggest a new way for designing highly efficient topological devices with compact bulk domains.

Interestingly, the newly developed hyperbolic band theory and crystallography of hyperbolic lattices[37–39] suggest that the hyperbolic lattices obeying discrete non-abelian translation groups can also possess a reciprocal-space description using generalized Bloch theorem. In this theory, the hyperbolic eigenstates are automorphic functions and the associated Brillouin zone is a higher-dimensional torus. Motivated by the hyperbolic band theory, the hyperbolic topological band insulators with non-trivial first Chern numbers have been theoretically created[40]. Moreover, hyperbolic graphene with the feature of topological semimetal has also been proposed[41]. While, up to now, the revealed hyperbolic band topologies are most related to the first Chern number. Generalization of the hyperbolic band topology with low-dimensional topological invariants to that with high-dimensional topological invariants is expected to introduce more novel effects. Hence, the question is whether hyperbolic topological states with non-zero $n$th Chern numbers exist, and how to realize those novel hyperbolic topological phases in experiments.

In this work, we report the first experimental observation of hyperbolic band topology with non-trivial second Chern numbers in electric circuit networks. Our model possesses the translational symmetry of a {8,8} hyperbolic tiling, and the corresponding momentum space is 4D. By engineering intercell couplings and onsite potentials of the hyperbolic model, topological bandgaps with non-zero second Chern numbers appear. The effectiveness of hyperbolic band theory with discretized crystal momentums is further confirmed by the consistence of calculated eigen-spectra to that based on the direct diagonalization. In experiments, we fabricate two types of hyperbolic circuits with periodic boundary conditions (PBCs) and partially open boundary conditions (OBCs) to demonstrate hyperbolic band topological states protected by second Chern numbers. By recovering the circuit admittance spectra and measuring impedance responses of hyperbolic circuits with PBCs, non-trivial bandgaps are clearly illustrated. Moreover, the topological boundary states are observed in hyperbolic circuits with partially OBCs, where the significant boundary impedance peaks appear in topological bandgaps. Furthermore, the measured impedance distributions are also matched to profiles of topological boundary states of hyperbolic models. Our work suggests a new method to engineer hyperbolic topological states with higher-order topological invariants, and gives an opportunity to explore much novel topological states in non-Euclidean spaces.

## Results

### The theory of hyperbolic band topology with second Chern numbers

We start to design a tight-binding lattice model in the 2D hyperbolic space. Figure 1a illustrates the Bravais lattice of designed hyperbolic model in a Poincaré disk, where the translational symmetry of a {8,8} hyperbolic tiling exists. The {8,8} hyperbolic lattice corresponds to the tessellation of the Poincaré disk by octagons, and each site has a coordination number of eight. Each unit cell (enclosed by the pink block), which is the fundamental tile in 2D hyperbolic space, contains four sublattice sites, as marked by colored dots in bottom inset. The onsite potentials of these sublattices equal to $m - a$, $m + a$, $-m + a$ and $-m - a$, respectively. These quantities are called as mass terms in the following. The infinite hyperbolic model can be constructed by tiling the 2D hyperbolic space with the unit cell along eight translational directions, which are marked by colored arrows labeling from $\gamma_1$ to $\gamma_4^{-1}$. The inter-cell coupling patterns along these directions are illustrated in the

four subplots of Fig. 1b, where different coupling strengths [$\pm J_j$ ($j$ = 1, 2, 3, 4), $\pm t_j$ ($j$ = 1, 3), and $\pm it_j$ ($j$ = 2, 4)] are represented by different types of solid and dashed lines.

Following the hyperbolic band theory, our proposed lattice model in 2D hyperbolic space can be described in the momentum space. In particular, we can equip inter-cell couplings of the hyperbolic unit cell with twisted boundary conditions along four translation directions, where the $U(1)$ phase factors $e^{ik_j}$ ($j$ = 1, 2, 3, 4) along directions given by $\gamma_1$, $\gamma_2$, $\gamma_3$, and $\gamma_4$ are introduced, as shown in Fig. 1a. The phase factors related to their inverses are in the form of $e^{-ik_j}$ ($j$ = 1, 2, 3, 4). In this case, four wave-vectors $k_1, k_2, k_3$, and $k_4$ can be regarded as Bloch vectors in 4D momentum space. Thus, we can apply Bloch's theorem in such a momentum space, making the corresponding Brillouin zone (BZ) become 4D. The hyperbolic Bloch Hamiltonian of our tight-binding lattice model can be expressed as $H = \boldsymbol{d}(\boldsymbol{k}) \cdot \boldsymbol{\Gamma} + ia\Gamma_1\Gamma_4$, where the vector $\boldsymbol{d}(\boldsymbol{k})$ is in the form of $\boldsymbol{d}(\boldsymbol{k}) = \{t_1 \sin(k_1), t_2 \sin(k_2), t_3 \sin(k_3), t_4 \sin(k_4), m + \sum_{j=1}^{4} J_j \cos(k_j)\}$ and the vector of gamma matrices $\boldsymbol{\Gamma} = \{\Gamma_1, \Gamma_2, \Gamma_3, \Gamma_4, \Gamma_5\}$ satisfies the Clifford algebra. Detailed expressions of these matrices are written as $\Gamma_1 = -\sigma_2 \otimes I$, $\Gamma_2 = \sigma_1 \otimes \sigma_1$, $\Gamma_3 = \sigma_1 \otimes \sigma_2$, $\Gamma_4 = \sigma_1 \otimes \sigma_3$, and $\Gamma_5 = \sigma_3 \otimes I$. $I$ is the 2 by 2 identity matrix and $\sigma_j$ ($j$ = 1, 2, 3) are Pauli matrices. See Supplementary Note 1 for the detailed deviation of $H(k)$ by the Fourier transform of real space hyperbolic Hamiltonian.

Figure 1c–e present the calculated hyperbolic energy bands at $k_1 = k_4 = 0$ with three different mass terms ($m = 0$, $a = 0$), ($m = 0.7$, $a = 0.2$) and ($m = 0.7$, $a = 3.2$), respectively. The value of $t_j$ and $J_j$ ($j$ = 1, 2, 3, 4) always equal to 1. It can be seen that Dirac points appear at $\varepsilon = 0$ with the mass term being zero ($m = 0$, $a = 0$). Those Dirac points are protected by the existence of time-reversal symmetry and inversion symmetry. It is noted that the non-zero values of $m$ and $a$ can break the inversion and time-reversal symmetries, respectively. In this case, by introducing the non-zero mass term with ($m = 0.7$, $a = 0.2$), the Dirac points are gapped, as shown in Fig. 1d. We can define the second Chern number in the 4D momentum space (corresponding to four translation directions) of 2D hyperbolic model as $C_2 = \frac{1}{8\pi^2} \int d^4k \ tr(\Omega_- \wedge \Omega_-)$, where the integral is taken over the 4D BZ of the {8,8} hyperbolic tiling and the trace runs over the wedge product of Berry curvature ($\Omega_-$) for all energy bands lower than the Fermi energy. In this case, we find that the opened bandgap (the shaded region) possesses a non-trivial second Chern number with $C_2 = 3$ but a vanishing first Chern number. In addition, by further increasing the value of $a$ ($m = 0.7$, $a = 3.2$), the topological bandgap around $\varepsilon = 0$ is closed, and other two bandgaps around $\varepsilon = \pm 3.2$ appear, as shown in Fig. 1e. We also calculate the second Chern numbers of these two bandgaps, and find that they all equal to zero. While, the non-zero first Chern number defined in 2D BZ formed by the momentum pair $(k_1, k_4)$ equals to $C_1(k_1, k_4) = -1$ ($C_1(k_1, k_4) = 1$) for the energy gap around $= -3.2$ ($\varepsilon = 3.2$). From above results, we can see that by suitably tuning mass terms, our proposed lattice model in the 2D hyperbolic space can exhibit topological physics with non-trivial second Chern numbers. In addition, we note that our proposed hyperbolic Bloch Hamiltonian with $a = 0$ has the same form with that of 4D quantum Hall model in Euclidean space[42], indicating that exotic topological physics should also exist in the finite hyperbolic model.

To explore the nontrivial hyperbolic topological physics with nonzero second Chern numbers in a finite lattice, we firstly impose PBCs to the hyperbolic model with twelve units, which are enclosed by pink circles in Fig. 1a and are labeled from '$a$' to '$l$'. It is important to note that there are many different ways for the realization of PBCs in a finite hyperbolic model. A recent study[39] has shown that the number of connection patterns for the 12-unit hyperbolic model with PBCs equals to that of distinct normal subgroups of index twelve for the Fuchsian group $\Gamma_{\{8,8\}}$ of the hyperbolic Bravais lattice {8,8}. Because of the non-abelian nature of hyperbolic translation group $\Gamma_{\{8,8\}}$, the high-dimensional representation may appear. In this case, different

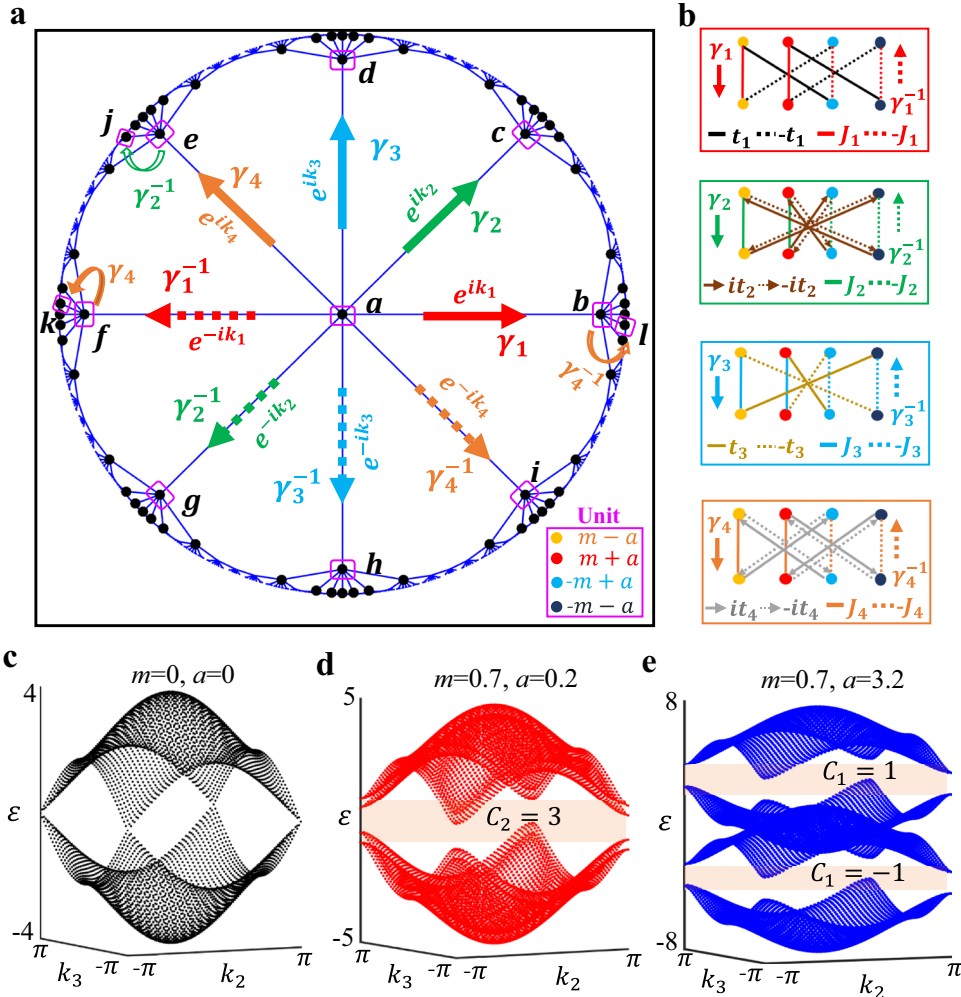

**Fig. 1 | The tight-binding lattice model in 2D hyperbolic space with second Chern numbers. a** The Bravais lattice {8,8} of proposed hyperbolic models in a Poincaré disk. The translation group of the designed hyperbolic model is a Fuchsian group defined by $\Gamma_{\{8,8\}} = <\gamma_1,\gamma_2,\gamma_3,\gamma_4,\gamma_1^{-1},\gamma_2^{-1},\gamma_3^{-1},\gamma_4^{-1} : \gamma_1\gamma_2^{-1}\gamma_3\gamma_4^{-1}\gamma_1^{-1}\gamma_2\gamma_3^{-1}\gamma_4>$, where $\gamma_1$ to $\gamma_4^{-1}$ manifest group generators of the hyperbolic Bravais lattice in a Poincaré disk and are marked by eight colored arrows. Dots filled with different colors in the bottom inset illustrate four sublattices in a single unit and the corresponding onsite potentials equal to $m-a$, $m+a$, $-m+a$, $-m-a$, respectively.

Pink circles labeled from '$a$' to '$l$' label twelve units of the finite hyperbolic lattice model. **b** The inter-cell coupling patterns of four sublattices along eight translation directions. Different types of solid and dashed lines correspond to different coupling strengths. **c–e** The calculated hyperbolic energy bands at $k_1 = k_4 = 0$ with three different mass terms ($m = 0$, $a = 0$), ($m = 0.7$, $a = 0.2$), and ($m = 0.7$, $a = 3.2$). The value of $t_j$ and $J_j$ ($j = 1, 2, 3, 4$) always equal to 1. The orange shaded regions correspond to non-trivial bandgaps with non-zero first or second Chern number.

boundary connections for PBCs could make the finite hyperbolic model obey or violate hyperbolic band theory. The finite lattice model with PBCs is called as the abelian (non-abelian) cluster when the hyperbolic band theory is satisfied (broken). Detailed descriptions on inter-cell couplings between boundary units of abelian and non-abelian clusters are illustrated in Supplementary Fig. 1 of Supplementary Note 2. Because, only abelian clusters satisfy the $U(1)$ hyperbolic Bloch band theory, which gives the nontrivial second Chern numbers in 2D hyperbolic space, in the following, we always focus on the abelian clusters.

Then, we perform a direct diagonalization of finite hyperbolic clusters with different mass terms, and numerical results are presented in Fig. 2a ($m = 0.7$, $a = 0.2$) and Fig. 2b ($m = 0.7$, $a = 3.2$) with blue circles. For comparison, we also explore hyperbolic band theory with discretized hyperbolic crystal momentums to calculate eigen-spectra of finite hyperbolic clusters with non-trivial topologies, as shown in Fig. 2a, b with red dots. See Supplementary Note 3 for detailed discussions on hyperbolic band theory of abelian clusters. We can see that there is an exact match between energy spectra computed by two different ways, indicating the effectiveness of hyperbolic band theory.

The locations of nontrivial bandgaps marked by orange shaded regions are consistent with that of $k$-space eigen-spectra. The trivial gaps marked by blue shaded regions are due to the finite discretization of hyperbolic crystal momentum. It is worthy to note that the significant difference exists for energy spectra of abelian and non-abelian hyperbolic clusters, manifesting that the non-abelian cluster could not be described by $U(1)$ hyperbolic band theory (see Supplementary Fig. 2 in Supplementary Note 4). To further illustrate the distribution of hyperbolic bulk modes, in Fig. 2c, d, we plot the spatial profiles of eigenmodes marked by arrows in Fig. 2a, b. Dashed blocks enclose four sublattices in unit cells labeled from '$a$' to '$l$'. We can see that extended bulk eigenmodes on a specific type of sublattices appear for abelian clusters with PBCs.

Next, to investigate the topological boundary states resulting from the non-zero second Chern number, the OBC should be introduced into the topological lattice model in 2D hyperbolic space. While, the full OBC could induce the appearance of a macroscopic fraction of boundary sites, making the definition of non-trivial bulk-energy gaps become fuzzy (see numerical results of Supplementary Fig. 3 in Supplementary Note 5). To overcome this obstacle, the OBC is only set on a

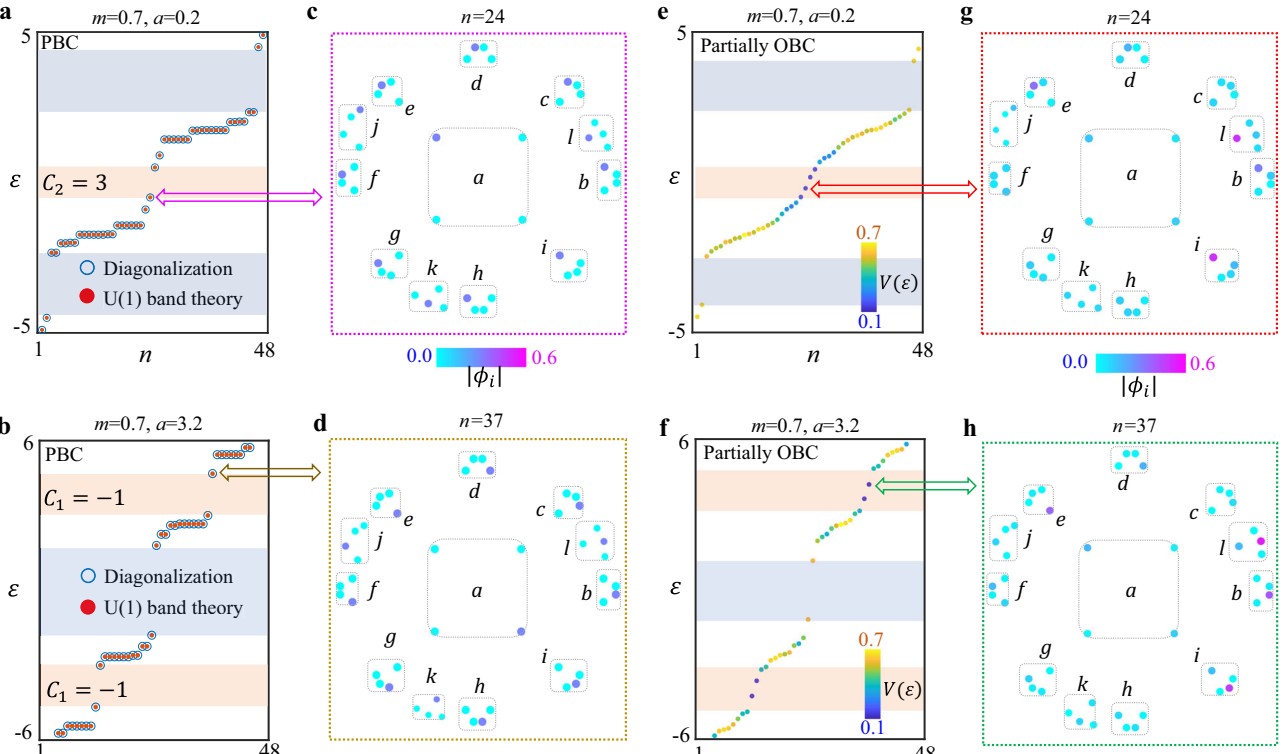

**Fig. 2 | Numerical results of finite hyperbolic clusters with PBCs and partially OBCs. a**, **b** The eigen-spectra of hyperbolic abelian clusters under PBCs, and the mass terms are ($m = 0.7$, $a = 0.2$) and ($m = 0.7$, $a = 3.2$). The blue circle and red dots correspond to numerical results obtained by the direct diagonalization and $U(1)$ hyperbolic band theory, respectively. **c**, **d** The spatial profiles of eigenmodes (marked by blue and orange arrows in Fig. 2a, b) for the topological hyperbolic clusters under PBCs with mass term being ($m = 0.7$, $a = 0.2$) and ($m = 0.7$, $a = 3.2$), respectively. **e**, **f** The eigen-spectra of hyperbolic abelian clusters under partially OBCs, and the mass terms are ($m = 0.7$, $a = 0.2$) and ($m = 0.7$, $a = 3.2$), respectively. The colormap corresponds to the quantity $V(\varepsilon)$. **g**, **h** The spatial profiles of eigen-modes (marked by red and green arrows in Fig. 2e, f) for the topological hyperbolic clusters with partial OBCs, and the mass terms are ($m = 0.7$, $a = 0.2$) and ($m = 0.7$, $a = 3.2$), respectively. Orange and blue shaded regions correspond to non-trivial and trivial bandgaps, respectively.

few boundary units in the hyperbolic cluster. In this case, we delete eight inter-cell couplings between boundary units marked by letters of '$i$', '$l$', '$e$', '$b$', '$c$', and '$d$'. In particular, open boundaries along five and four directions are introduced to units '$i$' and '$l$'. And, the open boundary condition along a single direction (two directions) is applied for '$b$' unit ('$e$','$c$', '$d$' units). Hence, '$i$','$l$', '$e$', '$b$', '$c$', and '$d$' are boundary units, where the corresponding coordination numbers of four sub-lattices in these units equal to 6, 8, 12, 14, 12, and 12, respectively (see Supplementary Fig. 4 in Supplementary Note 6 for details on partially OBCs). The coordination number of sublattices in other bulk units is 16. As shown in Fig. 2e, f, we numerically calculate the energy spectra of the hyperbolic cluster sustaining partial open boundaries with the mass term being ($m = 0.7$, $a = 0.2$) and ($m = 0.7$, $a = 3.2$), respectively. In addition, to quantify the localization degree of each eigenmode on boundary sites, a quantity $V(\varepsilon) = \sum_{i \in boundary} |\phi_i(\varepsilon)|^2 / \sum_i |\phi_i(\varepsilon)|^2$ is calculated (illustrated by the colormaps in Fig. 2e, f). $\phi_i(\varepsilon)$ is the probability amplitude at lattice site $i$ with the eigen-energy being $\varepsilon$. As expected, we find that the topological boundary states appear in nontrivial bandgaps possessing either non-zero first or second Chern number. In addition, the spatial distributions of topological boundary modes (marked by arrows in Fig. 2e, f) are displayed in Fig. 2g, h. Forty-eight points correspond to all lattice sites in the hyperbolic model with twelve units, where four sublattices in each unit are enclosed by dashed blocks. It is clearly shown that these midgap topological states exhibit the feature of significant boundary localizations around hyperbolic units sustaining open boundaries. In addition, it is important to note that, owing to different dimensions between momentum- and position-spaces of the hyperbolic model, the method for the three-dimensional cut through the four-dimensional Brillouin zone of {8,8}

hyperbolic lattice remains inconclusive. In this case, the way to introduce open boundaries along a fixed direction in the hyperbolic cluster is still to be explored, making the chiral propagation induced by topological invariants in the momentum space become hard to realize. While, we still find that midgap boundary states can only exist in bandgaps of hyperbolic clusters with non-trivial second Chern numbers. These properties are consistent with key features of topological boundary states induced by nontrivial Chern numbers.

In fact, the observation of theoretically predicted hyperbolic band topology with nontrivial second Chern numbers is not an easy task in real quantum and classical wave systems. In the next part, we construct 2D hyperbolic circuit networks to simulate those exotic topological phenomena.

### Experimental observation of hyperbolic band topology with second Chern numbers by artificial circuit networks

Motivated by recent experimental breakthroughs in realizing various quantum phases by electric circuits[43–52], in the following, we design hyperbolic circuit networks to observe the above proposed hyperbolic band topology with second Chern numbers. Figure 3a illustrates the photograph image of the fabricated circuit sample, which corresponds to the periodic hyperbolic cluster with twelve units (as illustrated in Fig. 1a). It is noted that several non-planar wire-crossings exist in our fabricated hyperbolic circuits. To realize those non-planar wire-crossings, the fabricated printed circuit board (PCB) possesses multilayers to arrange all planar and non-planar wire-crossings (see Methods for details). The enlarged view of the unit $a$ (enclosed by the pink circle) is plotted in the right chart. Specifically, four circuit nodes connected by capacitors $C$ (enclosed by the green

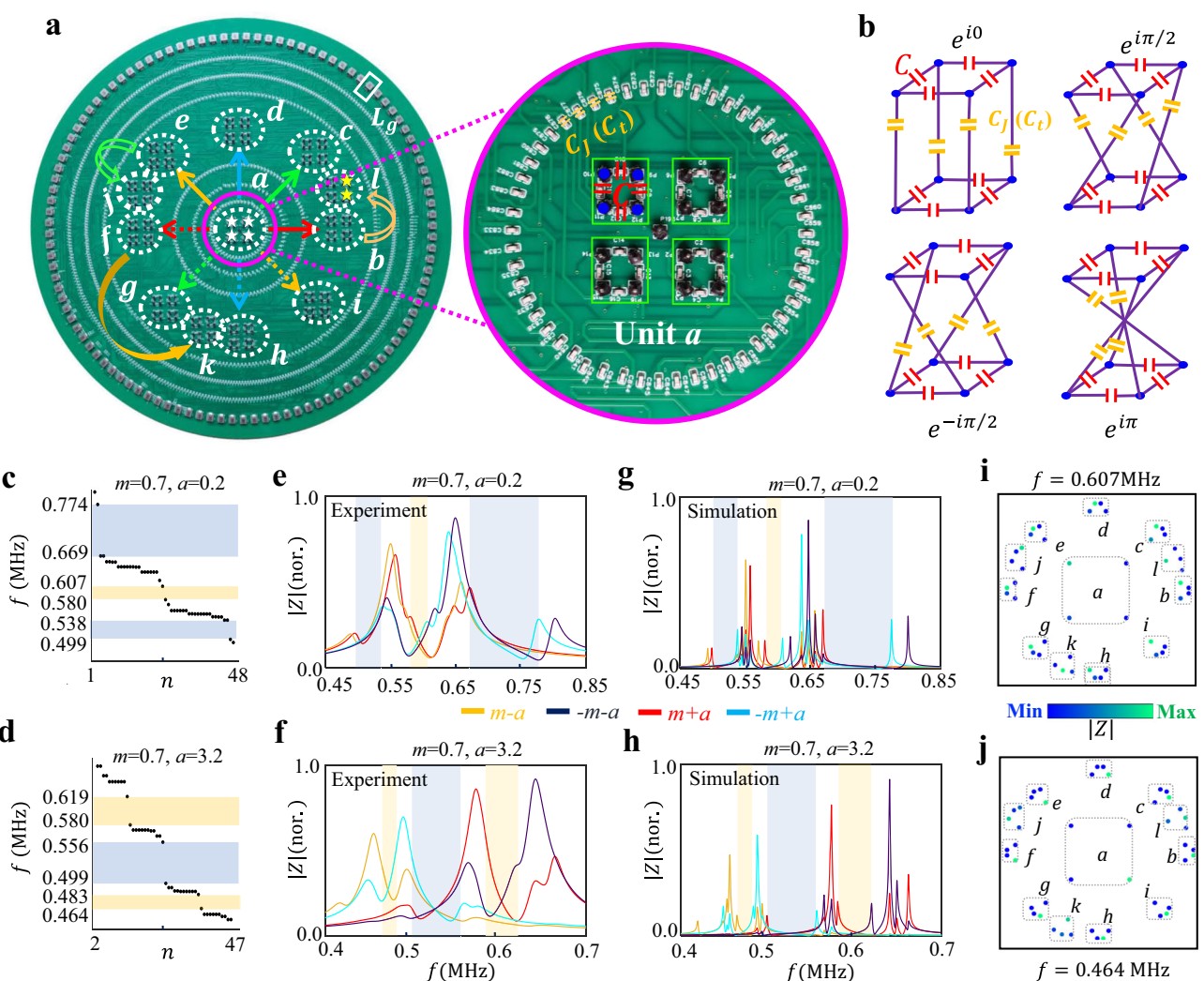

**Fig. 3 | Experimental results on hyperbolic band topology with second Chern numbers in artificial circuit networks with PBCs. a** The photograph image of the fabricated hyperbolic circuit sample under PBCs. Right charts present the front and back sides of enlarged views. White dash circles labeled from 'a' to 'l' label twelve units of the finite hyperbolic lattice model. Group generators $\gamma_1$ to $\gamma_4^{-1}$ are marked by eight colored arrows. **b** The schematic diagram for the realization of different inter-cell couplings. **c, d** The recovered circuit frequency spectra with the mass term being ($m = 0.7$, $a = 0.2$) and ($m = 0.7$, $a = 3.2$) under PBCs. **e, f** The measured impedance responses of four bulk nodes marked by four white stars in the 'a' unit cell with mass terms being ($m = 0.7$, $a = 0.2$) and ($m = 0.7$, $a = 3.2$), respectively. **g** and **h** Simulation results of frequency-dependent impedance responses with mass terms being ($m = 0.7$, $a = 0.2$) and ($m = 0.7$, $a = 3.2$). **i, j** Measured spatial impedance distributions at 0.607 MHz ($m = 0.7$, $a = 0.2$) and 0.464 MHz ($m = 0.7$, $a = 3.2$), respectively. Orange and blue shaded regions correspond to non-trivial and trivial bandgaps, respectively.

block) are considered to form an effective lattice site. Voltages on these four nodes are defined by $V_{i,1}$, $V_{i,2}$, $V_{i,3}$, and $V_{i,4}$, which could be suitably formulated to construct a pair of pseudospins ($V_{\uparrow i,\downarrow i} = V_{i,1} + V_{i,2}e^{\pm i\pi/2} + V_{i,3}e^{\pm i\pi} + V_{i,4}e^{\pm i3\pi/2}$) for realizing required site couplings. The schematic diagrams for the realization of different inter-cell couplings are shown in Fig. 3b. In particular, to simulate the real-valued hopping rate $J$ ($t$), four capacitors $C_J$ ($C_t$) are used to directly link adjacent nodes without a cross. For the realization of inter-site couplings with direction-dependent hopping phases $Je^{\pm i\pi/2}$ ($te^{\pm i\pi/2}$) and $Je^{i\pi}$ ($te^{i\pi}$), four pairs of adjacent nodes are connected crossly via four capacitors $C_J$ ($C_t$) in different ways. In addition, each circuit node is grounded by an inductor $L_g$. Four types of capacitors $C_1 = (m - a)C_g$, $C_2 = (m + a)C_g$, $C_3 = (-m + a)C_g$ and $C_4 = (-m - a)C_g$ are used for grounding on four sites in each unit to simulate the effective mass term. Moreover, to ensure the positive grounding capacitance on each node, an extra capacitor $C_u$ equaling to $(m + a)C_g$ is added to connect each circuit node to the ground.

Through the appropriate setting of grounding and connecting, the circuit eigenequation is identical to that of the hyperbolic tight-binding lattice model. Details for the derivation of circuit eigenequations are provided in Supplementary Note 7. In particular, the probability amplitude at the lattice site $i$ is mapped to the voltage of pseudospin $V_{\downarrow,i}$. Amplitudes of the effective inter-cell couplings equal to $t_j = -1$ and $J_j = -1$ with ($j = 1, 2, 3, 4$). The eigenenergy of hyperbolic lattice model is directly related to the eigenfrequency of circuit network as $\varepsilon = f_0^2/f^2 - 2 - (8C_J + 8C_t + C_u)/C$ with $f_0 = (2\pi\sqrt{CL_g})^{-1}$. It is noted that the tolerance of circuit elements is only 1% to avoid the detuning of circuit responses, and circuit parameters are set as $C = 2$ nF, $C_{J/t} = 1$ nF, $C_g = 2$ nF, and $L_g = 3.3$ uH.

To explore the topological states in hyperbolic circuit samples, we firstly recover the frequency spectra of circuit samples with different mass terms under PBCs (see Methods for details), as shown in Fig. 3c

(with $m = 0.7$, $a = 0.2$) and Fig. 3d (with $m = 0.7$, $a = 3.2$), respectively. We can see that the recovered frequency spectra of circuit possess identical amounts of bandgaps with respect to eigen-spectra calculated by hyperbolic band theory in Fig. 2a, b, giving the experimental demonstration of correctness for hyperbolic band theory. The difference of relative sizes of each bandgap results from the nonlinear relationship between $\varepsilon$ and $f$. It is known that impedance responses of different circuit nodes in the frequency-domain are related to the local density of states of corresponding quantum lattice models. In this case, we measure the frequency-dependent impedance response of four sublattices in a unit cell to illustrate the corresponding bulk spectrum. Measured results are plotted in Fig. 3e, f with mass terms being ($m = 0.7$, $a = 0.2$) and ($m = 0.7$, $a = 3.2$), respectively. It is shown that there is no impedance peak in frequency ranges of [0.499 MHz, 0.538 MHz], [0.58 MHz, 0.607 MHz], and [0.669 MHz, 0.774 MHz] for ($m = 0.7$, $a = 0.2$) and [0.464 MHz, 0.483 MHz], [0.499 MHz, 0.556 MHz], and [0.58 MHz, 0.619 MHz] for ($m = 0.7$, $a = 3.2$), respectively. The absence of impedance peaks at some frequency ranges (marked by shaded regions) is a direct evidence for the existence of bandgaps with PBCs. The frequency ranges of circuit bandgaps are consistent with energy gaps of the lattice model. Figure 3g, h presents the simulation results of frequency-dependent impedance responses of identical circuit nodes used in experiments by LTSPICE software. A good consistence between simulations and measurements is obtained, and the larger width of measured impedance peaks compared to simulation counterparts is originated from the large lossy effect in fabricated circuits (see Supplementary Fig. 5 in Supplementary Note 8). In addition, the spatial impedance distribution at 0.607 MHz (0.464 MHz) of the circuit sample with $m = 0.7$ and $a = 0.2$ ($m = 0.7$ and $a = 3.2$) is further measured, as presented in Fig. 3i (Fig. 3j). It is clearly shown that the spatial impedance profiles are matched to the corresponding extended bulk modes in Fig. 2c, d.

Then, to further investigate the topological properties of hyperbolic circuit samples, we turn to the circuit networks with partially OBCs, where the midgap boundary states resulting from non-trivial bulk topologies appear. Firstly, we consider the case with the mass term equaling to ($m = 0.7$, $a = 0.2$). The measured frequency-dependent impedance response of a boundary node, which corresponds to the sub-node with the effective onsite potential being $-m + a$ in the '$l$' unit, is shown in Fig. 4a with the green line. In addition, impedances of four bulk nodes in the '$a$' unit are also measured. It is seen that the disappearance of bulk impedance peak in three frequency ranges (marked by blue and orange regions) indicates the existence of bulk bandgaps. It is also found that there is a significant impedance peak of the boundary node in the frequency range of [0.58 MHz, 0.607 MHz] (marked by the shaded region in orange), which is matched to the topological bandgap with nonzero second Chern numbers for the hyperbolic cluster. Figure 4b presents the associated simulation results, which are consistent with measurements. To obtain the spatial distribution of topological boundary state induced by non-trivial second Chern numbers, we further measure the spatial impedance profile at 0.598 MHz, as shown in Fig. 4c. We can see that the measured impedance distribution is strongly concentrated around boundary units, which is consistent with the spatial profile of topological boundary mode plotted in Fig. 2g.

Next, we change the effective mass term to ($m = 0.7$, $a = 3.2$) and detect the topological boundary states induced by the first Chern number in hyperbolic circuits with partially OBCs. The measured frequency-dependent impedances of two boundary nodes, which correspond to sub-nodes with effective onsite potentials being $-m + a$ and $-m - a$ in the '$l$' unit, are shown by green and pink lines in Fig. 4d. Impedances of four bulk nodes in the '$a$' unit are also measured. It is shown that there are large impedance peaks of boundary nodes in the frequency ranges of [0.464 MHz, 0.483 MHz] and [0.58 MHz, 0.619 MHz], which are in accord with non-trivial energy gaps for the

hyperbolic cluster with nonzero first Chern numbers. The experimental results are also consistent with circuit simulations, as shown in Fig. 4e. Furthermore, in Fig. 4f, we measure the spatial impedance profile at 0.469 MHz, that is in a good consistent with the topological boundary state shown in Fig. 2h. Based on the above experimental results, it is clearly shown that the hyperbolic band topologies with non-trivial second and first Chern numbers have been successfully achieved in our designed artificial circuit networks.

## Discussion

We report the first experimental observation of hyperbolic band topology with non-trivial second Chern numbers in artificial circuit networks. Our designed hyperbolic model possesses the non-abelian translational symmetry of a {8,8} hyperbolic tiling, where the hyperbolic structure can be constructed by applying non-abelian translational operations generated by four generators of $\gamma_1$–$\gamma_4$ to different units. In this case, four generators can be mapped to four translational directions, inducing a 4D momentum space of the 2D hyperbolic lattice. By engineering intercell couplings and onsite potentials of four sublattices, topological bandgaps possessing non-trivial second Chern numbers can appear in the designed 2D hyperbolic model. To explore the nontrivial topological states in finite hyperbolic models, we firstly impose PBCs to construct abelian hyperbolic clusters with nontrivial bandgaps. The hyperbolic eigen-spectra calculated by direct diagonalizations are consistent with that evaluated by hyperbolic band theory. Then, we apply the partially OBCs to finite hyperbolic models, and find that the topological boundary states appear in nontrivial bandgaps with non-zero second Chern numbers. In experiments, we design and fabricate hyperbolic circuits with both PBCs and partially OBCs to detect the hyperbolic topological states. By recovering the circuit frequency spectra and measuring the impedance responses, the non-trivial bandgaps with either first or second Chern number and associated topological boundary states are clearly illustrated. In addition, the measured spatial impedance distributions are consistent with eigenmodes of the designed lattice model in 2D hyperbolic space. Furthermore, it is worth noting that there are no limitations on hyperbolic clusters listed in ref. [39] that can be implemented in circuit lattices, where any non-local and non-planar connections can be implemented by suitably designing circuit PCBs with multilayers. Moreover, there is no technical limitation that hinder further enlarging hyperbolic clusters. While, to ensure the large hyperbolic clusters to possess high-quality performances as their small-scale counterparts, the lossy effect in designed circuits as well as the influence of parasitic capacitances in large-scale PCBs should be carefully optimized. In this case, electric circuits are versatile platforms for emulating more exotic hyperbolic physics in future works.

Moreover, it is worth noting that our circuit implementation of hyperbolic physics possesses two breakthroughs compared to previous works on the construction of hyperbolic circuits. One is that the present work reports the first experimental realization of hyperbolic lattice models with periodic boundary conditions, where the designed and fabricated circuits have a highly non-planar and complicated network structure. Based on the hyperbolic circuit cluster with periodic boundary conditions, the hyperbolic hand theory has been experimentally demonstrated at the first time. In addition, the previously proposed boundary-dominated topological states in hyperbolic circuits are induced by the real space Chern numbers, and the associated circuit networks possess the fully open boundaries. While, our circuit firstly demonstrates the high-dimensional topology in 2D hyperbolic circuits, where the partially open boundary condition (not the fully open boundary condition) is a prerequisite. In this case, our work shows that the curvature can trigger the appearance of high-dimensional topological states in 2D systems. Such a phenomenon is the unique hyperbolic physics that has not been revealed in previous works.

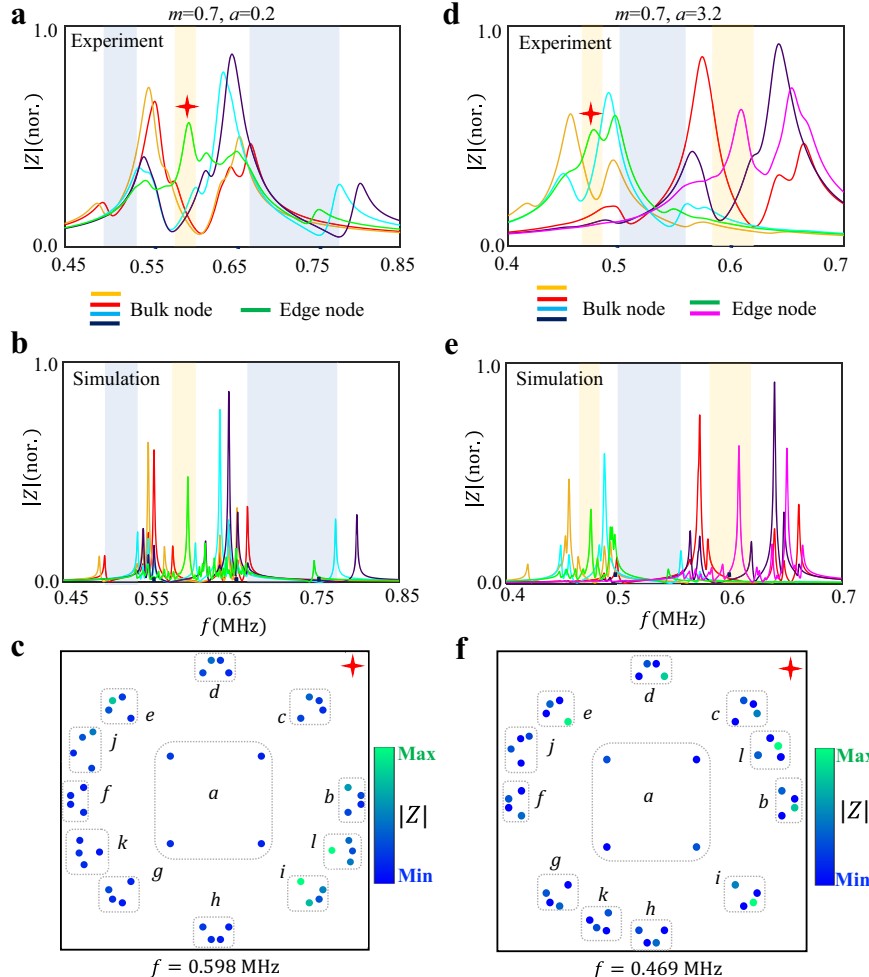

**Fig. 4 | Experimental results on hyperbolic topology edge states induced by second Chern numbers in artificial circuit networks with partially OBCs.**
**a, b** The measured and simulated impedance responses of bulk and boundary nodes in the hyperbolic circuit with the mass term being ($m = 0.7$, $a = 0.2$) under partially OBCs. Here, four bulk nodes correspond to four sublattices in '$a$' unit cell and the boundary node is the sub-node with the effective onsite potential being $-m + a$ in the '$l$' unit. **c** The measured spatial impedance profile at 0.598 MHz of the hyperbolic circuit with mass term being ($m = 0.7$, $a = 0.2$). **d, e** The measured and simulated impedance responses of bulk and boundary nodes in the hyperbolic

circuit with the mass term being ($m = 0.7$, $a = 3.2$) under partially OBCs. Here, four bulk nodes are consistent with that used in (**a, b**), and two boundary nodes correspond to sub-nodes with effective onsite potentials being $-m + a$ and $-m - a$ in the '$l$' unit. Bulk and boundary nodes are marked by white and yellow stars in Fig. 3a. **f** The spatial impedance profile at 0.469 MHz of the hyperbolic circuit with the mass term being ($m = 0.7$, $a = 3.2$) under partially OBCs. Red stars mark impedance peaks related to non-trivial boundary states. Orange and blue shaded regions correspond to non-trivial and trivial bandgaps, respectively.

Our work suggests a method to engineer topological states with 4D momentum spaces in 2D hyperbolic space. Different from previous experiments on the implementation of four-dimensional quantum Hall model with 2D topological charge pumps[53,54] and non-local circuit connections[55], the dimension enhancement of our proposed hyperbolic band topology protected by second Chern numbers is purely resulting from the unique property of Fuchsian translation group induced by the negative curvature. Except for the 4D quantum Hall physics, we could also investigate other 4D topological states, such as hexadecapole insulator and non-abelian Tensor monopoles, in 2D hyperbolic space by inversely designing the hyperbolic lattice model guided by the associated Euclidean Hamiltonian. In addition, it is known that even much higher dimensional momentum spaces (for example $\Gamma_{\{12,12\}}$ and $\Gamma_{\{16,16\}}$ correspond to six- and eight-dimensional momentum spaces) could be engineered in 2D hyperbolic space, paving a new way to explore much higher-dimensional band topologies in curved spaces.

## Methods

### Sample fabrications and circuit measurements

We exploit electric circuits by using LCEDA program software, where the PCB composition, stack-up layout, internal layer and grounding design are suitably engineered. Here, designed PCBs have six layers, where two layers are used for the inner electric layers and remaining four layers are used to arrange all planar and non-planar wire-crossings. Using the multilayer design of circuit PCBs, we note that electric circuits provide an ideal platform for embedding hyperbolic clusters with non-planar wire crossings on a flat physical geometry. It is worth noting that the all grounded components are grounded through blind buried holes. Moreover, all PCB traces have a relatively large width (0.75 mm) to reduce the parasitic inductance, and the spacing between electronic devices is also large enough to avert spurious inductive coupling. The SMP connectors are welded on the PCB nodes for the signal input. To ensure the tolerance of circuit elements and series resistance of inductors to be as low as possible, we use a WK6500B

impedance analyzer to select circuit elements with a high accuracy (the disorder strength is only 1%) and low losses.

To effectively excite hyperbolic circuits with complex couplings, four subnodes, which act as a single lattice site in the tight-binding lattice model, should be suitably excited with required phase differences. It is noted that eigen-spectra of hyperbolic circuits with respect to a pair of pseudospins $V_{\uparrow i} = V_{i,1} + V_{i,2}e^{i\pi/2} + V_{i,3}e^{i\pi} + V_{i,4}e^{i3\pi/2}$ and $V_{\downarrow i} = V_{i,1} + V_{i,2}e^{-i\pi/2} + V_{i,3}e^{i\pi} + V_{i,4}e^{-i3\pi/2}$ are degenerated. In this case, we can set the input signals at four circuit nodes as $(V_0, 0, -V_0, 0)$ to simultaneously excite these two pseudospins, where the measured impedance spectra and impedance profiles possess the same form by uniquely excite one of pseudospins [$V_{\uparrow i} = (V_0, iV_0, -V_0, -iV_0)$ or $V_{\downarrow i} = (V_0, -iV_0, -V_0, iV_0)$]. In addition, to recover the circuit frequency spectrum, the global voltage responses of all circuit nodes should be measured by lock-in amplifiers with a high dynamical range when the AC current with a constant amplitude is injected into a circuit node. Then, we repeat this procedure by exciting each circuit node. The current source can be realized by a voltage source connected to the PCB through a shunt resistance. Here, $V_i$ ($j$) corresponds to the measured voltage at node $i$ under the current excitation at node $j$ with $I_j$. Based on the measured voltages and input currents, we can get the inverse of circuit Laplacian $G$ ($G_{ij} = V_i$ ($j$)/$I_j$). In this case, the circuit Laplacian can be obtained as $J = G^{-1}$. Based on the recovered circuit Laplacian, the Hamiltonian of the associated hyperbolic lattice model can be written as $H = (\frac{1}{\omega^2 L_g C} - 2 - \frac{8C_J + 8C_t + C_u}{C})*\mathbb{I} - J$ with $\mathbb{I}$ being an identical matrix (see Supplementary note 6 for details). In this case, we can get the circuit frequency spectrum ($f = \omega/2\pi$) and the associated mode profiles with eigenvalues and eigenvectors of the recovered circuit Laplacian.

## Data availability
All data are displayed in the main text and Supplementary Information.

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

## Acknowledgements

This work is supported by the National Key R & D Program of China under Grant No. 2022YFA1404900 and National Science Foundation for Young Scientists of China No. 12104041.

## Author contributions

W.Z. finished the theoretical scheme and designed the circuit simulator. F.D. finished the experiments with the help of X.Z. and H.S. W.Z. and X.Z. wrote the paper. X.Z. initiated and designed this research project.

## Competing interests

The authors declare no competing interests.
