## [Peer Review File · Nature Communications]

Hyperbolic band topology with non-trivial second Chern numbersReviewer #1 (Remarks to the Author):

The manuscript reports on a combined theoretical and experimental study of higher-dimensional topology in band structures of hyperbolic lattices. A hyperbolic lattice of $\{p,q\}$ -type is a lattice of p -gonal faces with coordination number q . For the choice $p=q=8$ made here, the lattice is embedded into hyperbolic space of negative curvature. Such lattices can be realized in circuit quantum electrodynamics or, as in the present study, in classical electrical circuits. A striking feature of hyperbolic lattices is that momentum space is four- or higher-dimensional, such that 2nd, 3rd, etc. Chern numbers of Bloch bands in higher-dimensions are possible---in principle. Prior to this work, no topological hyperbolic lattice model with a nonvanishing 2nd Chern number was known or proposed. The present topic and realization thus strikes me as highly original both theoretically and experimentally.

The theoretical model introduced here is based on an $\{8,8\}$ -lattice with 4D momentum space, where four sites occupy each unit cell. The arguably engineered, but on the other hand not overly complicated, structure of real and imaginary hopping terms and on-site potentials yields a system that can be shown to have a non-trivial 2nd Chern number for certain choices of parameters. The authors use hyperbolic band theory to compute this topological invariant. The experimental realization is achieved with both periodic and open boundary conditions of the lattice. Hyperbolic lattice models with periodic boundary conditions yield highly non-planar, complicated networks. To my knowledge, the present work reports the first experimental realization of such a network in an actual electric circuit (using $12 \times 4 = 48$ lattice sites.). I find this achievement very impressive and it might open a new frontier in the study of hyperbolic lattice model in experiment. Using both the experiments with periodic and open boundary conditions, they confirm the theoretical predictions on the energy spectra of topological bands and measure the mid-gap states induced on the boundary through the bulk-boundary correspondence.

This manuscript presents a substantial achievement both in theory and experiment on topological models on hyperbolic lattices. The experimental realization of an actual hyperbolic network with periodic boundaries is particularly impressive. The identification of boundary-modes from topology in the spectrum in simulation and experiment is very convincing. I believe that this work will have considerable impact on the broad field of topological models and synthetic classical and quantum simulation, and inspire and guide upcoming works on simulations of negatively curved spaces. I strongly recommend this work for publication in Nature Communications after the following improvements on the main text have been implemented:

(1) The Chern number calculated for the parameters presented in Fig. 2d is $C=3$. Can the authors comment on the implications of this value on the number of (chiral) propagating boundary modes? Is this visible in experiment/simulation?

(2) I find it hard to read the plots of eigenvectors in Figs 2 g,h for open boundary conditions. Can the authors indicate which sites are boundary sites in the plot? What is the point of these plots?

(3) In conjunction with this, in lines 176,177, the authors indicate which bonds are removed to get partially open boundary conditions. Can the authors indicate how many boundary sites are induced by this, and what coordination number they have? Does this choice of boundary sites lead to a connected path along the boundary? This seems to be a prerequisite for the propagation of boundary modes.

(4) There is no discussion in the main text of how the lattice with periodic boundary conditions is realized in the circuit shown in Fig 3a. I am assuming it has several non-planar wire-crossings. Or is the hyperbolic cluster chosen planar? If not, how are the wire-crossing implemented on the circuit board?

(5) Related to (4), I think the discussion section needs to contain some perspective on what kind of hyperbolic clusters (listed in Ref. 11) are possible with the present technique. Is the 12-site cluster chosen here special in any sense, and why was it chosen? How big can the clusters be chosen in principle? If the experimental implementation of non-planar lattices is not feasible, can they be implemented with the simulation software LTSPICE used here, in principle? Looking forward, I feel these questions are pressing for the field and should be clarified here.

Reviewer #2 (Remarks to the Author):

This manuscript theoretically and experimentally reports the existence of a nontrivial bandgap with second Chern number in the hyperbolic lattice. The article is generally well written, although most of the mathematical derivations and technical details are provided in the Supplemental Material. I suggest the authors provide more details in experiment. All in all, this article is a nice work and I am happy to recommend a publication after the authors address the comments below.

1. It is known that the second Chern number is defined in the 4D, while this model is a 2D hyperbolic lattice. The author should give some details about how the corresponding momentum space changes from 2D to 4D. I think that this is the key to justify if the second Chern number exists or not. I suggest that the supplementary can be considered for placement in Fig.1.

2. The authors should provide a detailed derivation to obtain the Hamiltonian in line 103 of the text? In my opinion, this Hamiltonian of the momentum space is not very straightforward and can be obtained by the Fourier transforming the real-space Hamiltonian.

3. The 10-th article of the reference has error that the author should change "Phys. Rev. B 105, 125008" to "Phys. Rev. B 105, 125118". Perhaps there are other errors in the references that the authors should carefully check and correct.

4. The structure of this {8,8} hyperbolic lattice, i.e., eight octagons are placed around each node, possesses the non-abelian translational symmetry is special. The author should explain in detail how this hyperbolic lattice is constructed, as in their previous work Nat. Commun. 13, 2937 (2022).

5. The point of this work is that the hyperbolic lattice with the non-abelian translational symmetry can induce the second Chern bandgap. The author should explain their connection.

6. In Figs. 3c and 3d, the description in the Method is the admittance spectrum, but it seems that the authors give the frequency spectrum. The transformation from admittance to frequency spectrum is lacking?

7. The experimental result of Fig. 4f is not very convincing, since the edge state has some distribution in the bulk nodes. Perhaps this is due to the finite-size effect. The author should increase the layers of the hyperbolic lattice and remeasure the impedance profile at 0.469 MHz or 0.6 MHz which looks much better?

8. In Figs. 4a-b and Figs. 4d-e, why are the simulated peaks of the bulk mode and the edge modes sharper than the experimental results? Is there an inevitable loss in the experiment? The author should explain it.

9. Are the two different Chen insulator boundary states both chiral? The authors can measure the voltage propagation to see if the boundary state is chiral.

10. In the line 351, the author stated "the voltages at all circuit nodes are measured at the same time" but without clarifying how it was done.

11. The specific point "a bulk node" (In the line 239) and "a boundary node" (In the line 287) should be marked in the photograph image of the fabricated hyperbolic circuit sample.

Reviewer #3 (Remarks to the Author):

Please find my review attached.

Reviewer #3 Attachment on the following page

The authors of the present paper report the realization of the topological state on a hyperbolic lattice. This state is characterized by a nontrivial second Chern number and its implementation has been carried out on an electric circuit network. Theoretically, they use the recently established hyperbolic Bloch band theory to study the topological state on the $\{8,8\}$ tessellation of the hyperbolic lattice. The specialty of the hyperbolic lies in the non-Abelian nature of the corresponding lattice translations, which then as a consequence yields a higher-dimensional ($d>2$) Brillouin zone. The existence of the states with a nontrivial second Chern number in this framework therefore does not come as a surprise, e.g. standard tenfold way periodic table feature higher-Chern number phases in $d>3$, as also the authors mentioned), given that their existence is a consequence of a non-Abelian fiber bundle on a Brillouin zone. In the present case, the existence of such a fiber bundle is facilitated by the noncommuting lattice translations and higher-dimensional Brillouin zone.

The main new result in this paper is the realization of such in a topoelectric circuit. This required some rather sophisticated circuit engineering to implement the desired hopping elements (Fig. 1 in the manuscript) from the hyperbolic lattice, with the implementation as shown in Fig. 3 in the manuscript. The experimental results are shown in Fig. 4 together with the results in numerical computations of the band structure. The results are backed up by Supplemental Materials.

All in all, these are very impressive achievements, but, in spite of this, I am not convinced that this paper is suitable for publication in Nature Communications.

My main concern about this paper is the lack of novelty to satisfy the high bar imposed by Nature Communications.

1. The reports of the circuit network realizations of hyperbolic topological states have already been carried out, see refs. 29, 39, 41.
2. As already mentioned, the existence of the topological states characterized by the second Chern number is not surprising in hyperbolic lattices.
3. I fail to find what is a fundamental difference between the boundary states in the second-Chern number topological states in the Euclidean and the hyperbolic lattices, except that in the later case these accumulate at the boundary, as a consequence of the underlying hyperbolic geometric.

Therefore, I cannot recommend this paper for publication in Nature Communications. It is more appropriate to be published in a more specialized journal.

Response Letter to Reviewers

We are grateful for the constructive comments on this manuscript (NCOMMS-22-40653-T) from all three reviewers.

In the text below, reviewer comments are quoted in blue and followed by our detailed response. We have also revised the manuscript and Supplemental Materials based on reviewer comments, and these updates are highlighted in red in those files. In the text below, these updates are also highlighted in *Italics*.

Response to comments of the reviewer #1

The manuscript reports on a combined theoretical and experimental study of higher-dimensional topology in band structures of hyperbolic lattices. A hyperbolic lattice of $\{p, q\}$ -type is a lattice of p -gonal faces with coordination number q . For the choice $p=q=8$ made here, the lattice is embedded into hyperbolic space of negative curvature. Such lattices can be realized in circuit quantum electrodynamics or, as in the present study, in classical electrical circuits. A striking feature of hyperbolic lattices is that momentum space is four- or higher-dimensional, such that 2nd, 3rd, etc. Chern numbers of Bloch bands in higher-dimensions are possible---in principle. Prior to this work, no topological hyperbolic lattice model with a non-vanishing 2nd Chern number was known or proposed. The present topic and realization thus strike me as highly original both theoretically and experimentally.

The theoretical model introduced here is based on an $\{8,8\}$ -lattice with 4D momentum space, where four sites occupy each unit cell. The arguably engineered, but on the other hand not overly complicated, structure of real and imaginary hopping terms and on-site potentials yields a system that can be shown to have a non-trivial 2nd Chern number for certain choices of parameters. The authors use hyperbolic band theory to compute this topological invariant. The experimental realization is achieved with both periodic and open boundary conditions of the lattice. Hyperbolic lattice models with periodic boundary conditions yield highly non-planar, complicated networks. To my knowledge, the present work reports the first experimental realization of such a network in an actual electric circuit (using $12 \times 4 = 48$ lattice sites). I find this achievement very impressive and it might open a new frontier in the study of hyperbolic lattice model in experiment. Using both the experiments with periodic and open boundary conditions, they confirm the theoretical predictions on the energy spectra of topological bands and measure the mid-gap states induced on the boundary through the bulk-boundary correspondence.

This manuscript presents a substantial achievement both in theory and experiment on topological models on hyperbolic lattices. The experimental realization of an actual hyperbolic network with periodic boundaries is particularly impressive. The identification of boundary-modes from topology in the spectrum in simulation and experiment is very convincing. I believe that this work will have considerable impact on the broad field of topological models and synthetic classical and quantum simulation, and inspire and guide upcoming works on simulations of negatively curved spaces. I strongly recommend this work for publication in Nature Communications after the following improvements on the main text have been implemented:

Reply: We would like to thank the reviewer for the high evaluation of our work and insightful

suggestions. In the following, we give detailed responses to all points proposed by the reviewer and implement them in our revised manuscript.

(1) The Chern number calculated for the parameters presented in Fig. 2d is $C=3$. Can the authors comment on the implications of this value on the number of (chiral) propagating boundary modes? Is this visible in experiment/simulation?

Reply: We would like to thank the reviewer for the useful comment. Based on the bulk-boundary correspondence in Euclidean space, it has been demonstrated that the second Chern number with $C=3$ can ensure the existence of totally three chiral boundary states at $3D$ boundaries of the $4D$ topological insulator, where the open boundary condition is applied along a fixed direction and periodic boundary conditions are applied along other three directions. Thus, the corresponding eigenspectrum of the $3D$ periodic superlattice can provide a clear evidence on the existence of three midgap chiral boundary states.

While, the method for the three-dimensional cut through the four-dimensional Brillouin zone of $\{8,8\}$ hyperbolic lattice remains inconclusive. This is resulting from the different dimensions between momentum- and position-spaces of hyperbolic lattices. In this case, the way to introduce open boundaries along a fixed direction in the hyperbolic cluster is still unknown. Although, it is hard to introduce open boundaries along a fixed direction and quantify the number of boundary modes, we still find that the midgap boundary states can only exist in bandgaps of hyperbolic clusters with non-trivial second/first Chern numbers.

Action taken:

- In page 8 of the revised manuscript, we have added the following discussion to illustrate the difference of dimensions between the momentum space and the position space of our hyperbolic model: *“In addition, it is important to note that, owing to different dimensions between momentum- and position-spaces of the hyperbolic model, the method for the three-dimensional cut through the four-dimensional Brillouin zone of $\{8,8\}$ hyperbolic lattice remains inconclusive. In this case, the way to introduce open boundaries along a fixed direction in the hyperbolic cluster is still to be explored, making the chiral propagation induced by topological invariants in the momentum space become hard to realize. While, we still find that midgap boundary states can only exist in bandgaps of hyperbolic clusters with non-trivial second Chern numbers.”.*

(2) I find it hard to read the plots of eigenvectors in Figs 2g, h for open boundary conditions. Can the authors indicate which sites are boundary sites in the plot? What is the point of these plots? (3) In conjunction with this, in lines 176,177, the authors indicate which bonds are removed to get partially open boundary conditions. Can the authors indicate how many boundary sites are induced by this, and what coordination number they have? Does this choice of boundary sites lead to a connected path along the boundary? This seems to be a prerequisite for the propagation of boundary modes.

Reply: We would like to thank the reviewer for the useful comment. The hyperbolic cluster with twelve units, which are labeled from ‘a’ to ‘l’ (shown in Fig. 1a of the main text), is considered in our work. Each unit possesses four sublattices and connects with eight units to realize periodic

boundary conditions. Fig. R1 illustrates the scheme of intercell connections of each unit to create the (Abelian) hyperbolic cluster.

To introduce partially open boundaries, we delete eight inter-cell connections between hyperbolic units marked by letters of ‘ i ’, ‘ l ’, ‘ e ’, ‘ b ’, ‘ c ’, and ‘ d ’. The deleted intercell couplings are highlighted in red in Fig. R1. In particular, open boundaries along five and four directions are introduced to units ‘ i ’ and ‘ l ’. And, open boundaries along one and two directions are introduced into ‘ b ’ and ‘ e ’, ‘ c ’, ‘ d ’ units, respectively. Hence, ‘ i ’, ‘ l ’, ‘ e ’, ‘ b ’, ‘ c ’, and ‘ d ’ are boundary units, where the corresponding coordination numbers of four sublattices in these units equal to 6, 8, 12, 14, 12 and 12, respectively. The 48 points in Figs. 2g and 2h correspond to all lattice sites in the hyperbolic model with twelve units, where four sublattices in each unit are enclosed by dashed blocks. We can see that the midgap eigenvector induced by the nontrivial second (first) Chern number plotted in Fig. 2g (Fig. 2h) possesses the significant boundary localization (around boundary units labeled by ‘ i ’ and ‘ l ’).

γ_1	e to g	j to d	f to a	k to e	g to k	h to j	i to l	l to c
γ_2	e to f	j to e	f to l	k to b	g to a	h to k	i to j	l to d
γ_3	e to i	j to b	f to j	k to c	g to l	h to a	i to k	l to e
γ_4	e to h	j to c	f to k	k to d	g to j	h to l	i to a	l to b
γ_1^{-1}	e to k	j to h	f to b	k to g	g to e	h to d	i to c	l to i
γ_2^{-1}	e to j	j to i	f to e	k to h	g to d	h to c	i to b	l to f
γ_3^{-1}	e to l	j to f	f to d	k to i	g to c	h to b	i to e	l to g
γ_4^{-1}	e to a	j to d	f to c	k to f	g to b	h to e	i to d	l to h

Fig. R1. Details for the partially open boundary conditions of Abelian hyperbolic clusters.

In addition, the connected paths of boundary units (together with bulk units) exist in the hyperbolic cluster with open boundaries (for example $e \rightarrow j \rightarrow i \rightarrow k \rightarrow e$). While, the dimensions of position- and momentum-spaces of hyperbolic clusters are different, making the chiral propagation of hyperbolic edge states is hard to appear. While, the midgap boundary states are still existing in hyperbolic bandgaps with non-trivial Chern numbers.

Action taken:

- In page 7 of the revised manuscript, we have added the following discussion to clarify the partially open boundary conditions: “*In particular, open boundaries along five and four directions are introduced to units ‘ i ’ and ‘ l ’. And, the open boundary condition along a single direction (two directions) is applied for ‘ b ’ unit (‘ e ’, ‘ c ’, ‘ d ’ units). Hence, ‘ i ’, ‘ l ’, ‘ e ’, ‘ b ’, ‘ c ’, and ‘ d ’ are boundary units, where the corresponding coordination numbers of four sublattices in these units equal to 6, 8, 12, 14, 12 and 12, respectively (see Supplementary Note 5 for details on partially OBCs). The coordination number of sublattices in other bulk units is 16.*”.
- In page 8 of the revised manuscript, we have added the following discussion to illustrate the eigenvectors in Figs. 2g and 2h: “*Forty-eight points correspond to all lattice sites in the hyperbolic model with twelve units, where four sublattices in each unit are enclosed by dashed blocks.*”.

(4) There is no discussion in the main text of how the lattice with periodic boundary conditions is

realized in the circuit shown in Fig 3a. I am assuming it has several non-planar wire-crossings. Or is the hyperbolic cluster chosen planar? If not, how are the wire-crossing implemented on the circuit board?

Reply: We would like to thank the reviewer for the useful comment. It is true that several non-planar wire-crossings exist in our fabricated hyperbolic circuits with PBCs and partially OBCs. Actually, to realize those non-planar wire-crossings, our designed printed circuit boards (PCBs) possess six layers, where two layers correspond to the inner electric layers and the remaining four layers are used to arrange all planar and non-planar wire-crossings.

Action taken:

- In page 8 of the revised manuscript, we have added the following discussion to illustrate the method for realizing non-planar wire-crossings in the hyperbolic circuit: *“It is noted that several non-planar wire-crossings exist in our fabricated hyperbolic circuits. To realize those non-planar wire-crossings, the fabricated printed circuit board (PCB) possesses multilayers to arrange all planar and non-planar wire-crossings (see Methods for details).”*.
- In Methods of the revised manuscript, we have added the following discussion to illustrate the multilayer design of circuit PCB: *“Here, designed PCBs have six layers, where two layers are used for the inner electric layers and remaining four layers are used to arrange all planar and non-planar wire-crossings. Using the multilayer design of circuit PCBs, we note that electric circuits provide an ideal platform for embedding hyperbolic clusters with non-planar wire crossings on a flat physical geometry.”*.

(5) Related to (4), I think the discussion section needs to contain some perspective on what kind of hyperbolic clusters (listed in Ref. 11) are possible with the present technique. Is the 12-site cluster chosen here special in any sense, and why was it chosen? How big can the clusters be chosen in principle? If the experimental implementation of non-planar lattices is not feasible, can they be implemented with the simulation software LTSPICE used here, in principle? Looking forward, I feel these questions are pressing for the field and should be clarified here.

Reply: We would like to thank the reviewer for the useful suggestion. We want to clarify that there are no limitations on hyperbolic clusters listed in Ref. 11 that can be implemented in circuit lattices and simulated in LTSPICE. It is noted that the circuit response only depends on the connection pattern among all nodes. In this case, any non-local and non-planar connections can be implemented by suitably designing circuit PCBs with multilayers, which can arrange all required long-range and non-planar couplings of hyperbolic clusters. Moreover, there is no technical limitation that hinder further enlarging hyperbolic clusters in future works (by increasing the size and the layer number of PCBs). While, to ensure large hyperbolic clusters to possess high-quality performances as their small-scale counterparts, the lossy effect in designed circuits as well as the influence of parasitic capacitances in large-scale PCBs should be carefully optimized. For example, circuit elements with high Q-factors can be used in large-scale circuits to decrease losses. And the spatial profile of circuit-node distributions can be suitably optimized to make the required node connections become easy to implemented and the associated parasitic capacitances can be minimized. Hence, we note that electric circuits are versatile platforms for emulating exotic hyperbolic physics in the future.

Action taken:

- In the discussion section of the revised manuscript, we have added the following discussion to illustrate the perspective and limitation on realizing hyperbolic clusters in electric circuits:
“Furthermore, it is worth noting that there are no limitations on hyperbolic clusters listed in Ref. 11 that can be implemented in circuit lattices, where any non-local and non-planar connections can be realized by suitably designing circuit PCBs with multilayers. Moreover, there is no technical limitation that hinder further enlarging hyperbolic clusters. While, to ensure the large hyperbolic clusters to possess high-quality performances as their small-scale counterparts, the lossy effect in designed circuits as well as the influence of parasitic capacitances in large-scale PCBs should be carefully optimized. In this case, electric circuits are versatile platforms for emulating more exotic hyperbolic physics in future works.”

Response to comments of reviewer #2

This manuscript theoretically and experimentally reports the existence of a nontrivial bandgap with second Chern number in the hyperbolic lattice. The article is generally well written, although most of the mathematical derivations and technical details are provided in the Supplemental Material. I suggest the authors provide more details in experiment. All in all, this article is a nice work and I am happy to recommend a publication after the authors address the comments below.

Reply: We would like to thank the reviewer for the high evaluation of our work and insightful suggestions. In the following, we give detailed responses to all points proposed by the reviewer and implement them in our revised manuscript.

1. It is known that the second Chern number is defined in the 4D, while this model is a 2D hyperbolic lattice. The author should give some details about how the corresponding momentum space changes from 2D to 4D. I think that this is the key to justify if the second Chern number exists or not. I suggest that the supplementary can be considered for placement in Fig.1.

Reply: We would like to thank the reviewer for the useful suggestion. Based on the crystallography of the {8,8} hyperbolic lattice (Ref. 10), it has been pointed out that the {8,8} hyperbolic lattice can be constructed by applying hyperbolic translational operations, which are generated by four fundamental group generators $\gamma_1, \gamma_2, \gamma_3$ and γ_4 , to a single unit cell. In this case, based on the hyperbolic band theory (Refs. 9 and 11), we can employ twisted boundary conditions to the hyperbolic unit cell along four translation directions (related to $\gamma_1, \gamma_2, \gamma_3$ and γ_4) to construct the Bloch Hamiltonian. As for the hyperbolic Bloch Hamiltonian, the hopping term between two sublattices in two unit cells, which are connected by a translation operator γ_j or γ_j^{-1} ($j = 1, 2, 3, 4$), should carry an extra phase term e^{ik_j} or e^{-ik_j} ($j=1, 2, 3, 4$), making the hyperbolic momentum space become 4D. In this case, the 4D hyperbolic energy bands can be computed and the second Chern number can exist in our designed 2D hyperbolic lattice model.

Action taken:

- In the revised Fig. 1a of the main text, we have added the illustration of Bloch phase terms e^{ik_j} (e^{-ik_j}) related to hyperbolic group generators γ_j (γ_j^{-1}) with $j = 1, 2, 3, 4$ to clarify the 4D momentum space of 2D hyperbolic lattice.
- In page 4 of the revised manuscript, we have added the following discussion to illustrate the 4D momentum space of 2D hyperbolic lattice in Fig. 1: *“In particular, we can equip inter-cell couplings of the hyperbolic unit cell with twisted boundary conditions along four translation directions, where the $U(1)$ phase factors e^{ik_j} ($j=1, 2, 3, 4$) along directions given by $\gamma_1, \gamma_2, \gamma_3$ and γ_4 are introduced, as shown in Fig. 1a. The phase factors related to their inverses are in the form of e^{-ik_j} ($j=1, 2, 3, 4$). In this case, four wave-vectors k_1, k_2, k_3 and k_4 can be regarded as Bloch vectors in 4D momentum space. Thus, we can apply Bloch’s theorem in such a momentum space, making the corresponding Brillouin zone (BZ) become 4D.”*.
- In the revised Supplementary Note 2, we have provided a detailed illustration on the process for generating the {8, 8} lattice model by four hyperbolic translational operations labeled by $\gamma_1, \gamma_2, \gamma_3$ and γ_4 .

2. The authors should provide a detailed derivation to obtain the Hamiltonian in line 103 of the text? In my opinion, this Hamiltonian of the momentum space is not very straightforward and can be obtained by the Fourier transforming the real-space Hamiltonian.

Reply: We would like to thank the reviewer for the useful suggestion. We agree with the reviewer that the hyperbolic Hamiltonian in momentum space can be obtained by performing Fourier transform on the real space Hamiltonian. In the revised Supplementary Note 1, we have provided a detailed derivation of $H(k)$ by this method.

Action taken:

- In the revised Supplementary Note 1, we have provided a detailed derivation of $H(k)$ by the Fourier transform of real space hyperbolic Hamiltonian.
- In page 4 of the revised manuscript, we have added the following discussion: “See Supplementary Note 1 for the detailed deviation of $H(k)$ by the Fourier transform of real space hyperbolic Hamiltonian.”.

3. The 10-th article of the reference has error that the author should change “Phys. Rev. B 105, 125008” to “Phys. Rev. B 105, 125118”. Perhaps there are other errors in the references that the authors should carefully check and correct.

Reply: We would like to thank the reviewer for pointing out this error. We have changed “Phys. Rev. B 105, 125008” to “Phys. Rev. B 105, 125118”, and also carefully checked other references.

4. The structure of this {8,8} hyperbolic lattice, i.e., eight octagons are placed around each node, possesses the non-abelian translational symmetry is special. The author should explain in detail how this hyperbolic lattice is constructed, as in their previous work Nat. Commun. 13, 2937 (2022).

Reply: We would like to thank the reviewer for the useful suggestion. The hyperbolic lattices we consider are of the $\{p, q\}$ type, which means that they are tessellations of the Poincaré disk by p -gons and each lattice site has the coordination number q . In this case, our considered {8,8} hyperbolic lattice corresponds to the tessellation of the Poincaré disk by octagons, and each site has the coordination number of eight.

Based on the crystallography of the {8,8} hyperbolic lattice (Ref. 10), it has been pointed out that the hyperbolic lattice can be constructed by applying hyperbolic translational operations to a single unit cell. In the following, we illustrate such a process for generating our designed hyperbolic lattice model. The representation of translational generators in the Poincaré disk ($\gamma_1, \gamma_2, \gamma_3$ and γ_4) can be expressed as $\gamma_u = R((u-1)\alpha_B)\gamma_1R(-(u-1)\alpha_B)$ with $u = 1, 2, 3, 4$. Here, R is the rotation matrix written by $R((u-1)\alpha_B) = \begin{pmatrix} \exp(i(u-1)\alpha_B/2) & \sigma \\ \sigma & \exp(i(u-1)\alpha_B/2) \end{pmatrix}$ with $\alpha_B = 2\pi/8$. γ_1 is the first generator described by $\gamma_1 = \frac{1}{\sqrt{1-\sigma^2}} \begin{pmatrix} 1 & \sigma \\ \sigma & 1 \end{pmatrix}$ with $\sigma = \sqrt{(\cos(\alpha_B) + \cos(\beta_B))/(1 + \cos(\beta_B))}$ and $\beta_B = 2\pi/8$. Performing these generators to the 2D coordinate of an operated unit cell, the hyperbolic model can be created.

As shown in Fig. R2a, we start to apply eight translational operations (γ_i and γ_i^{-1} with $i = 1, 2, 3, 4$) to the central unit with four sublattices, which are enclosed by the black dash block. These

operations on the central unit can generate eight first-generation units. And, the inter-cell couplings along these directions (not shown here) are illustrated in Fig. 1b of main text. Then, we apply eight translational operations to each first-generation unit (enclosed by red dash blocks), as shown in Fig. R2b. It is noted that there are eight second-generation units being coincident to the central unit. In this case, we have totally 56 second-generation units. Repeating the above process, we can obtain the hyperbolic model with high-generation units.

Fig. R2. The illustration of generating process for the hyperbolic lattice model.

Action taken:

- In the revised Supplementary Note 2, we have provided a detailed illustration of the process for generating the designed $\{8, 8\}$ hyperbolic lattice model.
- In page 3 of the revised manuscript, we have added the following discussion to illustrate the construction of the $\{8,8\}$ hyperbolic lattice: “*The $\{8,8\}$ hyperbolic lattice corresponds to the tessellation of the Poincaré disk by octagons, and each site has a coordination number of eight.*”.

5. The point of this work is that the hyperbolic lattice with the non-abelian translational symmetry can induce the second Chern bandgap. The author should explain their connection.

Reply: We would like to thank the reviewer for the useful comment. The crystallography of the $\{8,8\}$ hyperbolic lattice indicates that the $\{8,8\}$ hyperbolic lattice can be constructed by applying hyperbolic translational operations generated by four generators of $\gamma_1, \gamma_2, \gamma_3$ and γ_4 to different units (as illustrated in the above comment). In this case, four generators can be mapped to four translational directions, inducing a 4D momentum space of the $\{8,8\}$ hyperbolic lattice. In this case, the 4D hyperbolic energy band can be computed and the second Chern number can exist in the suitably engineered 2D hyperbolic lattice.

Action taken:

- In the discussion part of our revised manuscript, we have provided the following explanation on the connection between the hyperbolic translational symmetry and second Chern bandgaps in our model: “*Our designed hyperbolic model possesses the non-abelian translational symmetry of a $\{8,8\}$ hyperbolic tiling, where the hyperbolic structure can be constructed by applying non-abelian translational operations generated by four generators of $\gamma_1, \gamma_2, \gamma_3$ and γ_4 to different units. In this case, four generators can be mapped to four translational directions, inducing a 4D momentum space of the 2D hyperbolic lattice. By engineering the intercell couplings and onsite potentials of four sublattices, topological*

bandgaps possessing non-trivial second Chern numbers can appear in the designed 2D hyperbolic model.”.

6. In Figs. 3c and 3d, the description in the Method is the admittance spectrum, but it seems that the authors give the frequency spectrum. The transformation from admittance to frequency spectrum is lacking?

Reply: We would like to thank the reviewer for the comment. We want to clarify that the admittance spectrum provided in Figs. 3c and 3d are indeed the eigenspectra of effective Hamiltonian of circuits (see Supplementary Note 6 for details), where the eigen-frequency is mapped to the eigen-energy of the hyperbolic lattice. To avoid the misunderstanding, we rename the admittance spectrum as the frequency spectrum of circuit in the revised manuscript.

7. The experimental result of Fig. 4f is not very convincing, since the edge state has some distribution in the bulk nodes. Perhaps this is due to the finite-size effect. The author should increase the layers of the hyperbolic lattice and remeasure the impedance profile at 0.469 MHz or 0.6 MHz which looks much better?

Reply: We would like to thank the reviewer for the useful comment. We note that the little amplitude at bulk nodes of edge states in Fig. 4f is resulting from the imperfect excitation of pseudospin pairs of $(V_0, 0, -V_0, 0)$ in the fabricated circuit, which is required to implement effective hoppings in our designed hyperbolic lattice model. In this case, we remeasure the impedance profile at 0.469MHz by precisely exciting each circuit node with pseudospins pairs $(V_0, 0, -V_0, 0)$, as shown in Fig. R3. We can see that the nearly zero amplitudes of bulk nodes appear for the edge state.

Fig. R3. The remeasured impedance profile at 0.469MHz.

Action taken:

- In Fig. 4f of the revised manuscript, we have provided the remeasured impedance profile.

8. In Figs. 4a-b and Figs. 4d-e, why are the simulated peaks of the bulk mode and the edge modes sharper than the experimental results? Is there an inevitable loss in the experiment? The author should explain it.

Reply: We would like to thank the reviewer for the comment. It is true that the wider peaks in experiments are resulting from the lossy effect. To illustrate this effect, we calculate the impedance responses with the effective series resistances of inductance being $50\text{ m}\Omega$, $150\text{ m}\Omega$, and $300\text{ m}\Omega$, as shown in Fig. R4. Simulation results in Figs. R4a and R4b correspond to the circuit considered in Figs. 3e and 4a in the main text. And, simulation results in Figs. R4c and R4d correspond to the circuit with parameters identical to Figs. 3f and 4b, respectively. It is shown that with the series resistances of inductance being increased, the impedance peaks of both bulk and boundary nodes are broadening. These results clearly prove that the wider impedance peaks in experiments are resulting from large losses in fabricated circuits.

Fig. R4. Calculated impedance responses of hyperbolic circuits with different effective series resistances of inductance. (a) and (c). The simulated impedance responses of bulk nodes in periodic circuit with different losses, and the mass terms is $(m=0.7, a=0.2)$ and $(m=0.7, a=3.2)$, respectively. (b) and (d). The simulated impedance responses of bulk and boundary nodes in circuits possessing different losses under partially OBCs, and the mass terms is $(m=0.7, a=0.2)$ and $(m=0.7, a=3.2)$, respectively.

Action taken:

- In page 10 of the revised manuscript, we have added the following discussion to clarify the wider peaks in experiments are resulting from the lossy effect in circuits: “A good consistence between simulations and measurements is obtained, and the larger width of measured impedance peaks compared to simulation counterparts is originated from the large lossy effect in fabricated circuits (see Supplementary Note 8 for details).”.

9. Are the two different Chen insulator boundary states both chiral? The authors can measure the voltage propagation to see if the boundary state is chiral.

Reply: We would like to thank the reviewer for the useful comment. Based on the bulk-boundary correspondence in Euclidean space, it is noted that the non-zero second Chern number can ensure the existence of chiral boundary states at $3D$ boundaries in the $4D$ topological insulator, where the open boundary condition is applied along one direction and the periodic boundary conditions are applied along other three directions.

While, the method for the three-dimensional cut through the four-dimensional Brillouin zone of the $\{8,8\}$ hyperbolic lattice remains inconclusive. This is resulting from different dimensions between the momentum- and position-spaces of hyperbolic lattices. In this case, the way to introduce open boundaries along a fixed direction in the hyperbolic cluster is still to be explored. While, although it is hard to introduce open boundaries along a fixed direction and observe the chiral propagation in real space, we still find that the midgap boundary states can only exist in bandgaps of hyperbolic clusters with non-trivial second/first Chern numbers.

Action taken:

- In page 8 of the revised manuscript, we have added the following discussion to illustrate the difference between momentum- and position-spaces of hyperbolic models: “In addition, it is important to note that, owing to different dimensions between momentum- and position-spaces of the hyperbolic model, the method for the three-dimensional cut through the four-dimensional Brillouin zone of $\{8,8\}$ hyperbolic lattice remains inconclusive. In this case, the way to introduce open boundaries along a fixed direction in the hyperbolic cluster is still to be explored, making the chiral propagation induced by topological invariants in the momentum space become hard to realize. While, we still find that midgap boundary states can only exist in bandgaps of hyperbolic clusters with non-trivial second Chern numbers.”.

10. In the line 351, the author stated “the voltages at all circuit nodes are measured at the same time” but without clarifying how it was done.

Reply: We would like to thank the reviewer for the comment. To recover the circuit frequency spectrum, the global voltage responses of all circuit nodes should be measured by lock-in amplifiers with a high dynamical range when the AC current with a constant amplitude is injected to a circuit node. Then, we repeat this procedure by exciting each circuit node. The current source can be realized by a voltage source connected to the PCB through a shunt resistance. Here, $V_i(j)$ corresponds to the measured voltage at node i under the current excitation at node j with I_j . Based on the measured voltages and input currents, we can get the inverse of circuit Laplacian G ($G_{ij}=V_i(j)/I_j$). In this case, the circuit Laplacian can also be obtained with $J=G^{-1}$. Based on the

recovered circuit Laplacian, the Hamiltonian of the associated hyperbolic lattice model can be written as $H = \left(\frac{1}{\omega^2 L_g C} - 2 - \frac{8C_J + 8C_t + C_u}{c}\right) * \mathbb{I} - J$ with \mathbb{I} being an identical matrix (see Supplementary note 6 for details). In this case, we can get the circuit frequency spectrum ($f = \omega/2\pi$) and the associated mode profiles from eigenvalues and eigenvectors of the recovered circuit Laplacian.

Action taken:

- The detailed procedure for recovering the circuit frequency spectrum is added in the Method section of revised manuscript as: *“In addition, to recover the circuit frequency spectrum, the global voltage responses of all circuit nodes should be measured by lock-in amplifiers with a high dynamical range when the AC current with a constant amplitude is injected into a circuit node. Then, we repeat this procedure by exciting each circuit node. The current source can be realized by a voltage source connected to the PCB through a shunt resistance. Here, $V_i(j)$ corresponds to the measured voltage at node i under the current excitation at node j with I_j . Based on the measured voltages and input currents, we can get the inverse of circuit Laplacian G ($G_{ij} = V_i(j)/I_j$). In this case, the circuit Laplacian can be obtained as $J = G^{-1}$. Based on the recovered circuit Laplacian, the Hamiltonian of the associated hyperbolic lattice model can be written as $H = \left(\frac{1}{\omega^2 L_g C} - 2 - \frac{8C_J + 8C_t + C_u}{c}\right) * \mathbb{I} - J$ with \mathbb{I} being an identical matrix (see Supplementary note 6 for details). In this case, we can get the circuit frequency spectrum ($f = \omega/2\pi$) and the associated mode profiles with eigenvalues and eigenvectors of the recovered circuit Laplacian.”.*

11. The specific point “a bulk node” (In the line 239) and “a boundary node” (In the line 287) should be marked in the photograph image of the fabricated hyperbolic circuit sample.

Reply: We would like to thank the reviewer for the comment. We note that ‘four bulk nodes’ correspond to four subnodes in the ‘ a ’ unit, and “a boundary node” corresponds to the sub-node with the effective onsite potential being $-m+a$ in the ‘ l ’ unit. We have marked these circuit nodes by white and yellow stars in the photograph image of the fabricated circuit sample.

Response to comments of reviewer #3

The authors of the present paper report the realization of the topological state on a hyperbolic lattice. This state is characterized by a nontrivial second Chern number and its implementation has been carried out on an electric circuit network. Theoretically, they use the recently established hyperbolic Bloch band theory to study the topological state on the {8,8} tessellation of the hyperbolic lattice. The specialty of the hyperbolic lies in the non-Abelian nature of the corresponding lattice translations, which then as a consequence yields a higher-dimensional ($d > 2$) Brillouin zone. The existence of the states with a nontrivial second Chern number in this framework therefore does not come as a surprise, e.g. standard tenfold way periodic table feature higher-Chern number phases in $d > 3$, as also the authors mentioned), given that their existence is a consequence of a non-Abelian fiber bundle on a Brillouin zone. In the present case, the existence of such a fiber bundle is facilitated by the non-commuting lattice translations and higher-dimensional Brillouin zone.

The main new result in this paper is the realization of such in a topoelectric circuit. This required some rather sophisticated circuit engineering to implement the desired hopping elements (Fig. 1 in the manuscript) from the hyperbolic lattice, with the implementation as shown in Fig. 3 in the manuscript. The experimental results are shown in Fig. 4 together with the results in numerical computations of the band structure. The results are backed up by Supplemental Materials.

All in all, these are very impressive achievements, but, in spite of this, I am not convinced that this paper is suitable for publication in Nature Communications. My main concern about this paper is the lack of novelty to satisfy the high bar imposed by Nature Communications.

1. The reports of the circuit network realizations of hyperbolic topological states have already been carried out, see refs. 29, 39, 41.
2. As already mentioned, the existence of the topological states characterized by the second Chern number is not surprising in hyperbolic lattices.
3. I fail to find what is a fundamental difference between the boundary states in the second-Chern number topological states in the Euclidean and the hyperbolic lattices, except that in the latter case these accumulate at the boundary, as a consequence of the underlying hyperbolic geometric.

Therefore, I cannot recommend this paper for publication in Nature Communications. It is more appropriate to be published in a more specialized journal.

Reply: We would like to thank the reviewer for the careful review of our manuscript. The reviewer finds that our manuscript contains impressive achievements, but he/she is not convinced that the associated novelty can reach to the high bar imposed by Nature Communications. In fact, our manuscript presents an important achievement on topological models of hyperbolic lattices, as pointed out by the first and second reviewers, and contains the enough novelty both in theory and experiment. In the following, we give detailed explanations on the novelty of our works from three aspects raised by the reviewer.

Firstly, although hyperbolic topological states have been carried out, our circuit implementation of hyperbolic topological physics possesses two breakthroughs compared with previous investigations (Refs. 29, 39, 41). One is that the present work reports the first experimental realization of hyperbolic lattice models with *periodic boundary conditions*. It is different from previous works on hyperbolic topological lattices with open boundaries. In the case, our designed and fabricated circuits have a highly non-planar and complicated network structure. Based on the hyperbolic circuit cluster with periodic boundary conditions, hyperbolic band theory has been

experimentally demonstrated at the first time. Additionally, the previously proposed boundary-dominated topological states in hyperbolic circuits are induced by *the real space Chern numbers*, and the associated circuit networks possess the fully open boundaries. While, our circuit firstly demonstrates the high-dimensional topology in 2D hyperbolic circuits, where the partially open boundary condition (not the fully open boundary condition) is a prerequisite. In this case, our work firstly proves that the negative curvature can trigger the appearance of high-dimensional topological states in 2D systems. Such a phenomenon is the unique hyperbolic physics that has not been revealed in previous works.

Secondly, the realization of the hyperbolic topological states characterized by the second Chern number is not an easy task. In theory, we must suitably design boundary connections to construct Abelian clusters to match the hyperbolic hand theory. Moreover, the partially open boundary conditions should be introduced to observe the midgap topological states, where fully open boundaries can destroy the high-dimensional topological states (see Figure 3 in Supplementary Materials). In experiments, the long-range boundary connections are highly non-planar in fabricated hyperbolic circuits. Thus, it requires a more precise PCB design, where the optimization of arrangements and connections for all circuit elements are needed. Such a complicated circuit network with a non-trivial hyperbolic band topology is firstly constructed in our work. Hence, the realization of hyperbolic topology with higher-dimensional topological invariants are more complicated than previous works on the hyperbolic topological state with real-space Chern numbers both in theory and experiment.

Finally, there are fundamental differences between topological boundary states induced by the second-Chern number in the Euclidean and the hyperbolic lattices. Except for the boundary-dominated features, the other perimetry difference is that the dimensions of momentum- and position-spaces of hyperbolic clusters are different. In this case, the method for the three-dimensional cut through the four-dimensional Brillouin zone of $\{8,8\}$ hyperbolic lattice remains inconclusive. While, although it is hard to introduce open boundaries along a fixed direction of hyperbolic clusters, we still find that the midgap boundary states can exist in bandgaps with non-trivial second Chern numbers, where the partially open boundary conditions are suitably applied to the system. Our works also show that there should be a more precise bulk-boundary correspondence in hyperbolic lattices, which promote future investigations on the relation between position- and momentum-spaces in the hyperbolic matter.

In the revised manuscript, we have added more discussions to further illustrate the novelty of our work, and hope that our efforts can convince the reviewer for the recommendation of publishing.

Action taken:

- In the discussion part of our revised manuscript, we have added the following discussion to further illustrate the novelty of our work: *“Moreover, it is worth noting that our circuit implementation of hyperbolic physics possesses two breakthroughs compared to previous works on the construction of hyperbolic circuits. One is that the present work reports the first experimental realization of hyperbolic lattice models with periodic boundary conditions, where the designed and fabricated circuits have a highly non-planar and complicated network structure. Based on the hyperbolic circuit cluster with periodic boundary conditions, the hyperbolic hand theory has been experimentally demonstrated at the first time. In addition, the previously proposed boundary-dominated topological states in hyperbolic circuits are induced by the real space Chern numbers, and the associated circuit networks possess the fully open*

boundaries. While, our circuit firstly demonstrates the high-dimensional topology in 2D hyperbolic circuits, where the partially open boundary condition (not the fully open boundary condition) is a prerequisite. In this case, our work shows that the curvature can trigger the appearance of high-dimensional topological states in 2D systems. Such a phenomenon is the unique hyperbolic physics that has not been revealed in previous works.”

Reviewer #1 (Remarks to the Author):

The authors have successfully answered my queries and provided additional material in the revised version of the manuscript.

As I have written earlier, I find this work very impressive and strongly recommend publication in Nature Communications. What I find particularly exciting is the realization of a periodic cluster of lattice sites, which represents a highly non-planar graph. The authors use six layers of circuit board to implement their graph. I would have actually liked to see this feature be discussed more prominently in the text, maybe with some figures, but it is certainly explained in the text of the current version. To my knowledge, such multilayer implementations of non-planar graphs have not been realized before, and non-planarity is commonly considered a bottleneck for circuit realizations.

Since hyperbolic lattices with periodic boundaries are necessarily highly non-planar, there is a significant advancement in the present work as compared to earlier references Nat. Commun. 13, 2937 (2022) and Nat. Commun. 13, 4373 (2022), which were limited to planar graphs. I would consider the present work as a technical breakthrough and I am curious to see whether the authors will be able to make even bigger clusters in the future, thereby opening a new experimental frontier for simulation of topological states and Hamiltonians.

Reviewer #2 (Remarks to the Author):

The author solved many problems raised by the three referees and made some refinements to the presentation of the results, thus improving the quality of the manuscript in the new iteration. I am now comfortable in recommending this paper for publication.

Reviewer #3 (Remarks to the Author):

Please see the attached report.

Reviewer #3 Attachment on the following page

I thank the authors for their detailed response to my previous report.

As I have already stated in my first report, this is indeed a very impressive work, and particularly its experimental side is highly nontrivial. In particular, the realization of the partial open boundary conditions in the hyperbolic lattice geometry, where there is mismatch of the dimensionality between the momentum space (equal to four) and the real space (equal to two), is the main result of this work, showing that indeed a version of the bulk-boundary correspondence should be operative in this case too. The authors have now highlighted this in the text, but I think they should put even more emphasis on this aspect of the work. So, the authors satisfactorily responded to the point 3 of my report.

As for the point 1, I agree that the experimental realization of the hyperbolic topological state with a nontrivial 2nd Chern number with periodic and partially open boundary conditions is a nontrivial achievement, and different than in the mentioned references.

As for point 2, I admit that the the realization of the topological phase characterized by the 2nd Chern number on the hyperbolic lattice has not been previously achieved, as also I mentioned before. Also, technically, this is rather daunting task, as shown by the authors. However, my concern here is rather conceptual, i.e., whether this result brings anything conceptually new in the field of topological states of matter. The authors did not address this issue in their response, but rather put the emphasis on the technical advancements in their work. My opinion is that conceptually this finding does not go much beyond the features of the 4D quantum Hall effect, which is precisely the state corresponding to the topological state on the hyperbolic lattice described and experimentally realized by the authors. Again, my take here would be that this is not enough to cut a high bar imposed by Nature Communications.

In summary, although the authors have to some extent satisfactorily responded to the issues raised in my previous report, I still cannot recommend the paper for publication in Nature Communications.

Response to comments of the reviewer #1

The authors have successfully answered my queries and provided additional material in the revised version of the manuscript.

As I have written earlier, I find this work very impressive and strongly recommend publication in Nature Communications. What I find particularly exciting is the realization of a periodic cluster of lattice sites, which represents a highly non-planar graph. The authors use six layers of circuit board to implement their graph. I would have actually liked to see this feature be discussed more prominently in the text, maybe with some figures, but it is certainly explained in the text of the current version. To my knowledge, such multilayer implementations of non-planar graphs have not been realized before, and non-planarity is commonly considered a bottleneck for circuit realizations. Since hyperbolic lattices with periodic boundaries are necessarily highly non-planar, there is a significant advancement in the present work as compared to earlier references Nat. Commun. 13, 2937 (2022) and Nat. Commun. 13, 4373 (2022), which were limited to planar graphs. I would consider the present work as a technical breakthrough and I am curious to see whether the authors will be able to make even bigger clusters in the future, thereby opening a new experimental frontier for simulation of topological states and Hamiltonians.

Reply: We would like to thank the reviewer for the high evaluation and acceptance of our manuscript.

Response to comments of reviewer #2

The author solved many problems raised by the three referees and made some refinements to the presentation of the results, thus improving the quality of the manuscript in the new iteration. I am now comfortable in recommending this paper for publication.

Reply: We would like to thank the reviewer for the acceptance of our revised manuscript.

Response to comments of reviewer #3

I thank the authors for their detailed response to my previous report.

As I have already stated in my first report, this is indeed a very impressive work, and particularly its experimental side is highly nontrivial. In particular, the realization of the partial open boundary conditions in the hyperbolic lattice geometry, where there is mismatch of the dimensionality between the momentum space (equal to four) and the real space (equal to two), is the main result of this work, showing that indeed a version of the bulk-boundary correspondence should be operative in this case too. The authors have now highlighted this in the text, but I think they should put even more emphasis on this aspect of the work. So, the authors satisfactorily responded to the point 3 of my report. As for the point 1, I agree that the experimental realization of the hyperbolic topological state with a nontrivial 2nd Chern number with periodic and partially open boundary conditions is a nontrivial achievement, and different than in the mentioned references. As for point 2, I admit that the realization of the topological phase characterized by the 2nd Chern number on the hyperbolic lattice has not been previously achieved, as also I mentioned before. Also, technically, this is rather daunting task, as shown by the authors. However, my concern here is rather conceptual, i.e., whether this result brings anything conceptually new in the field of topological states of matter. The authors did not address this issue in their response, but rather put the emphasis on the technical advancements in their work. My opinion is that conceptually this finding does not go much beyond the features of the 4D quantum Hall effect, which is precisely the state corresponding to the topological state on the hyperbolic lattice described and experimentally realized by the authors. Again, my take here would be that this is not enough to cut a high bar imposed by Nature Communications. In summary, although the authors have to some extent satisfactorily responded to the issues raised in my previous report, I still cannot recommend the paper for publication in Nature Communications.

Reply: We would like to thank the reviewer for agreeing that our manuscript is indeed very impressive, and particularly the experimental side is highly nontrivial. In addition, we want to stress that our work also contains the conceptual advance. In particular, we promote the topological band theory to non-Euclidean space with second Chern numbers. Such a topological state is indeed different from 4D quantum Hall effect, where the dimensions of momentum- and position-spaces of hyperbolic counterparts are different. We believe that pursuing a more general version of the bulk-boundary correspondence with respect to the hyperbolic topological band theory is extremely exciting, and more future investigations and publications are needed.